# Prevention of Lipotoxicity in Pancreatic Islets with Gammahydroxybutyrate

**DOI:** 10.3390/cells11030545

**Published:** 2022-02-04

**Authors:** Justin Hou Ming Yung, Lucy Shu Nga Yeung, Aleksandar Ivovic, Yao Fang Tan, Emelien Mariella Jentz, Battsetseg Batchuluun, Himaben Gohil, Michael B. Wheeler, Jamie W. Joseph, Adria Giacca, Mortimer Mamelak

**Affiliations:** 1Department of Physiology, University of Toronto, Toronto, ON M5S1A8, Canada; justin.yung@mail.utoronto.ca (J.H.M.Y.); lucy.yeung@mail.utoronto.ca (L.S.N.Y.); a.ivovic@mail.utoronto.ca (A.I.); yao.tan@utoronto.ca (Y.F.T.); battsetseg.batchuluun@gmail.com (B.B.); hima.gohil@mail.utoronto.ca (H.G.); michael.wheeler@utoronto.ca (M.B.W.); 2School of Pharmacy, University of Waterloo, Kitchener, ON N2G1C5, Canada; emjentz@uwaterloo.ca (E.M.J.); jamie.joseph@uwaterloo.ca (J.W.J.); 3Department of Medicine, University of Toronto, Toronto, ON M5S1A8, Canada; 4Institute of Medical Science, University of Toronto, Toronto, ON M5S1A8, Canada; 5Banting and Best Diabetes Centre, University of Toronto, Toronto General Hospital, Toronto, ON M5S12C4, Canada; 6Department of Psychiatry, University of Toronto, Toronto, ON M6A2E1, Canada

**Keywords:** oxidative stress, ß-cell, lipotoxicity, gamma-hydroxybutyrate (GHB), type 2 diabetes, NADPH

## Abstract

Oxidative stress caused by the exposure of pancreatic ß-cells to high levels of fatty acids impairs insulin secretion. This lipotoxicity is thought to play an important role in ß-cell failure in type 2 diabetes and can be prevented by antioxidants. Gamma-hydroxybutyrate (GHB), an endogenous antioxidant and energy source, has previously been shown to protect mice from streptozotocin and alloxan-induced diabetes; both compounds are generators of oxidative stress and yield models of type-1 diabetes. We sought to determine whether GHB could protect mouse islets from lipotoxicity caused by palmitate, a model relevant to type 2 diabetes. We found that GHB prevented the generation of palmitate-induced reactive oxygen species and the associated lipotoxic inhibition of glucose-stimulated insulin secretion while increasing the NADPH/NADP+ ratio. GHB may owe its antioxidant and insulin secretory effects to the formation of NADPH.

## 1. Introduction

Type 2 diabetes is characterized by insulin resistance and defective insulin secretion. Obesity is the major predisposing factor to type 2 diabetes and accounts for the excessive release of fatty acids from the expanded adipose tissue mass. High plasma levels of non-esterified fatty acids (NEFA) are known to cause insulin resistance, but the prolonged exposure of pancreatic ß-cells to high fatty acid levels also impairs the insulin-producing capacity of these cells by a process that likely involves the generation of oxidative stress [1]. ß-cell damage by fatty acids and the subsequent reduction in insulin secretion appears to be an early event in the pathogenesis of hyperglycemia in type 2 diabetes. There is now good evidence that antioxidants can prevent the development of oxidative stress in ß-cells in response to fatty acids and that this can maintain the insulin-producing capacity of these cells [1,2]. Previous work in our laboratory has demonstrated that lipotoxicity is caused by the activation of islet NADPH oxidase by long-chain fatty acids and the ensuing formation of the superoxide radical and aldehydes. Antioxidants such as *n*-acetylcysteine, taurine, and tempol were shown to prevent the toxic effects of this oxidative stress [1,2].

We now set out to determine whether gamma-hydroxybutyrate (GHB), an intermediary metabolite in all living cells, can also prevent the damage to pancreatic islet cells, which occurs naturally in diabetes in response to high levels of fatty acids [3,4]. Previous studies have revealed that GHB can prevent the development of diabetes in mice challenged by streptozotocin and alloxan, two toxic agents that are employed to develop models of type 1 diabetes and that are known to generate oxidative stress in pancreatic islets [5,6]. Based on its capacity to prevent tissue damage following exposure to ionizing radiation or high oxygen pressure, Laborit [7] proposed that GHB owed its antioxidant properties to the activation of the pentose phosphate shunt (PPS) and the formation of the reducing coenzyme NADPH. Taberner et al. [8] confirmed that this was, in fact, the case, and GHB’s antioxidant powers have been repeatedly demonstrated in the brain and other tissues, but it has never been clear just how GHB activates the shunt [3,4].

In this study, GHB’s protective actions were examined in an *in vitro* model of ß-cell lipotoxicity, which, as stated above, is relevant to type 2 diabetes. GHB was shown to block the formation of reactive oxygen species (ROS) and to prevent the reduction in glucose-stimulated insulin secretion (GSIS) produced by palmitate, the most common saturated NEFA.

GHB is on the market under the trade name Xyrem^®^ and has been used safely every night for many years by patients with narcolepsy [9]. GHB’s capacity to protect the insulin-producing ß-cells of the pancreas from the toxic effects of high plasma levels of fatty acids may open the door to further investigation of this or related compounds as adjuvants in the treatment of type 2 diabetes.

## 2. Materials and Methods

### 2.1. Animals

All procedures were approved by the Animal Care Committee of the University of Toronto (animal protocol number: 20011526, approved 30 March 2021). Islets of 11–13 weeks old male C57BL/6 mice (Jackson Laboratory 000664) were isolated via collagenase (Sigma Aldrich Burlington, MA, USA, Cat. #C9263) injection into the common bile duct as previously described [2,10,11,12].

All mice were co-housed in cages containing 2–4 mice, and had access to water and standard chow (Envigo Teklad Global 18% Protein Rodent Diet (2918), Madison, WI, USA) ad libitum. Mice were not fasted prior to euthanasia and sample collection. Euthanasia was achieved by an overdose (0.2 mL/mouse) of ketamine:xylazine:acepromazine (150 mg/kg:5 mg/kg:1 mg/kg) cocktail injected intraperitoneally in isofluorane anesthetized mice. Islet isolation then commenced.

### 2.2. Islet Treatment In Vitro

After 1 hr recovery in RPMI1640 (without antioxidants) with 7% fetal bovine serum (FBS) containing 10 mM HEPES, freshly isolated islets from C57BL/6 mice were incubated for 48 hrs at 37 °C and 5% CO_2_ with: (1) control islet medium; (2) palmitate (0.4 mM); (3) palmitate (0.4 mM) and GHB (5 mM) or (4) control and GHB (5 mM). 48 hrs exposure of islets to this concentration of palmitate results in impairment in insulin secretion in islets [1]. The concentration of GHB was determined based on previous literature [3,8]. Health Canada exemption number for GHB: 51650.03.21.

### 2.3. Glucose-Stimulated Insulin Secretion (GSIS) Assay

Following islet treatment *in vitro*, islets were incubated in Krebs Ringer buffer containing 10 mM HEPES (KRBH) with 2.8 mM glucose for 1 hr for recovery. GSIS was then evaluated by incubation of islets in KRBH containing: 6.5 mM (basal glucose concentration for mice) or 22 mM glucose (maximum stimulatory concentration) in duplicate for 2 hrs. Supernatant insulin was measured with Linco’s RIA kit (EMB Millipore, Billercia, MA, USA, Cat. #RI-13K).

### 2.4. Reactive Oxygen Species (ROS) Measurement

ROS were detected with dihydro-dichlorofluorescein diacetate (H2DCF-DA) (Invitrogen, Waltham, MA, USA, Cat. #C6827). Following islet treatment *in vitro*, islets were incubated with 10 µM of H2DCF-DA in 2.8 mM glucose in KRBH for 20 min. After washing with KRBH, fluorescence was measured at 490 nm excitation and 510 nm emission. Ten islets were measured per treatment.

### 2.5. Oxygen Consumption Rate

Oxygen consumption rate was measured with Seahorse XF cell Mito Stress Test (Seahorse Bioscience, Billerica, MA, USA) according to manufacturer’s instructions. Prior to analysis, islets were preincubated in 525 μL of XF Base Medium (Seahorse Bioscience) containing 2 mM glutamine, 1 mM sodium pyruvate and 16.5 mM glucose at 37 °C without CO_2_. Four separate compounds were then injected sequentially: (1) glucose (final concentration 22 mM); (2) oligomycin, an ATP synthase inhibitor, (final concentration 5 μM); (3) FCCP, an uncoupling agent, (final concentration 2 μM); and (4) antimycin (final concentration 5 μM) with rotenone (final concentration 5 μM), a cellular respiration and electron transport chain inhibitor. Five OCR measurements were taken after each injection.

### 2.6. NADPH and NADP+ Measurement

Islet NADPH and NADP+ levels were measured with Sigma Aldrich NADP+/NADPH Assay Kit (Sigma Aldrich, Burlington, MA, USA, Cat. #MAK312) according to the manufacturer’s instructions. Prior to analysis, islets were collected and washed twice in 2.8 mM glucose in KRBH. After washing with KRBH, islets were frozen using a dry ice-ethanol bath for at least 5 min. Samples were then stored at −80 °C until analysis.

### 2.7. Statistical Analysis

Data are presented as means ± SE. Shapiro–Wilk tests on Prism 9 (La Jolla, CA, USA) were run first to determine normality of data. Nonparametric or parametric analyses were performed accordingly. One-way nonparametric analysis of variance (ANOVA on ranks) or parametric ANOVA for repeated measurements (when appropriate) followed by Tukey’s test was used to compare differences between treatments. Calculations were performed using SAS (Cary, NC, USA).

## 3. Results

Figure 1 illustrates the protective effect of 5 mM GHB on insulin secretion in this *in vitro* model. Palmitate significantly decreased GSIS as expected. Despite exposure to palmitate, islets concomitantly treated with GHB maintained normal levels of insulin secretion. GHB alone did not affect insulin secretion.

Figure 2a,b illustrate the significant increase in ROS generated in mouse islets treated with palmitate and the absence of this effect in islets concomitantly treated with GHB.

Figure 3 reveals that palmitate tended to reduce islet oxygen consumption and that this appeared to be prevented by GHB. GHB alone also tended to decrease oxygen consumption. However, none of these effects were statistically significant.

Figure 4a reveals that palmitate tended to reduce islet NADPH levels while co-treatment of palmitate with GHB significantly increased NADPH levels compared to islets treated only with palmitate (*p* < 0.01). Figure 4b reveals that palmitate tended to increase islet NADP+ levels. Lastly, Figure 4c shows that palmitate reduced the ratio of NADPH/NADP+, which was prevented by co-treatment with GHB (*p* < 0.05 vs. ALL, *p* < 0.01 vs. PAL + GHB and GHB).

## 4. Discussion

Exposure of pancreatic ß-cells to NEFA induces the formation of ROS and impairs GSIS, a common measure of ß-cell function [1,2]. Pancreatic ß-cells have reduced antioxidant enzyme gene expression compared to other cells and correspondingly low levels of free radical detoxifying and redox regulating enzymes, which may account for their unusual vulnerability to ROS-induced oxidative stress [11]. Oxidative stress decreases insulin secretion by various mechanisms such as the induction of endoplasmic reticulum stress and inflammation, and the reduction of mitochondrial function and ultimately cell viability [13].

While ROS may be generated along the mitochondrial electron transport chain in the course of fatty acid oxidation or as a result of mitochondrial membrane damage, the major source of ROS generated in pancreatic ß-cells in response to NEFA appears to be the formation of superoxide by the cytoplasmic/plasma membrane NADPH oxidase complex. NEFA have been shown to increase the activity of NADPH oxidase in islets [14,15], in part by PKC activation [16].

Our pharmacological and genetic studies confirm the role of NADPH oxidase in the generation of ROS by NEFA and the role of these ROS in the impairment of ß-cell function *in vivo* [2]. NADPH is a substrate for NADPH oxidase, and its oxidation should reduce the NADPH/NADP+ ratio in the cell, as found in the present study. However, measurements indicate that NEFA may not affect or actually increase the intracellular NADPH/NADP+ ratio [2,17,18]. In addition to the PPS, there are two other major sources of NADPH. A mitochondrial enzyme, nicotinamide nucleotide transhydrogenase, can form NADPH from NADH; however, this enzyme is absent in this strain of C57BL/6 mice from Jackson [19]. NADPH can also be formed from cytosolic malate as described below when acetyl-CoA in mitochondria produced in the course of NEFA oxidation allosterically activates pyruvate carboxylase to form oxaloacetate. This important anaplerotic reaction is followed by the conversion of oxaloacetate to malate by malate dehydrogenase. Malate then enters the cytoplasm, where it is decarboxylated to pyruvate by the malic enzyme together with the formation of NADPH. This malate-pyruvate shunt can produce far more NADPH than the PPS, which is known to be a very minor route of glucose metabolism in pancreatic ß-cells. Glucose and succinate, which are both precursors of pyruvate and malate, can also activate the shunt and generate NADPH. However, the prolonged exposure of ß-cells to NEFA suppresses pyruvate carboxylase gene transcription and the NADPH-generating activities of the malate pyruvate shunt [20], which may further explain the decrease in the NADPH/NADP+ ratio seen with palmitate in the present study.

One important function of NADPH in the ß-cell is to serve as a cofactor for antioxidant enzymes, including glutathione reductase, thiol reductase, and quinone reductase. Decreased formation of NADPH thus elevates the level of ROS. The rise in the level of ROS may then uncouple the electron transport chain, interfere with the synthesis of ATP and insulin secretion, and ultimately threaten the viability of the ß-cells [18].

The NADPH/NADP+ ratio also regulates the response of the ß-cells to two major insulin secretagogues, glucose and succinate. Both of these secretagogues enhance the formation of NADPH. The flow of electrons from NADPH to glutathione and to the redox proteins glutaredoxin and thioredoxin mediates insulin granule exocytosis [21]. Insulin release is also regulated by the NADPH/NADP+ ratio’s effect on Kv channels. Active Kv channels repolarize the ß-cells. A high intracellular NADPH/NADP+ ratio inactivates these Kv channels and thus increases insulin secretion by maintaining membrane depolarization [22].

Our findings indicate that GHB is effective at blocking the formation of ROS and preventing the inhibitory effect of the NEFA palmitate on GSIS. GHB’s antioxidant powers have been attributed to its capacity to activate the PPS and generate NADPH [6,7,8,23]. In mice, *in vivo*, high intraperitoneal doses of GHB (500 mg/kg) increase the ratio of 1-14C/6-14C in expired air, a measure of the shift in glucose metabolism to the PPS, by 300%. The same effect is seen with slices of cerebral cortical grey matter in rats but not with slices of kidney or diaphragm. Intraperitoneal doses of 500 mg/kg in mice and rats increase glucose-6-phosphate dehydrogenase activity in the whole brain by 27%. However, the activity of glucose-6-phosphate dehydrogenase *in vitro* is not altered by high concentrations of GHB [8]. Taberner et al. [8] were not able to explain how GHB activated the PPS and generated NADPH, and, to our knowledge, the reason for this activation has never been satisfactorily explained. Glucose-6-phosphate dehydrogenase, a cytosolic enzyme, is the rate-limiting step in the PPS and is not known to be activated by GHB and, in fact, is negatively regulated by an increasing NADPH/NADP+ ratio. Utilization of NADPH and the ensuing rise in the level of NADP+ activates the enzyme [24].

GHB is metabolized in both the cytoplasm and mitochondria [25,26]. An NADP+ dependent oxidoreductase oxidizes GHB in the cytoplasm to succinic semialdehyde with the concomitant formation of NADPH. This reaction is inhibited by NADPH and proceeds slowly but it is greatly accelerated in the brain, liver, and kidney when it is coupled to the reduction of D-glucuronate to L-gulonate by NADPH and the regeneration of NADP+. The coupled reaction results in the rapid formation of succinic semialdehyde without any overall change in the levels of either NADPH or NADP+ [25,26]. However, this has not been investigated in the ß-cell. In mitochondria, a hydroxyacid-oxoacid transhydrogenase catalyzes the transformation of GHB and α-ketoglutarate to succinic semialdehyde and D-α-hydroxyglutarate. In both cases, succinic semialdehyde is rapidly converted to succinic acid with the concomitant reduction of NAD+ to NADH [25,26].

In our *in vitro* experimental set-up, islets were incubated for 48 h in the presence of palmitate with or without GHB. ROS formation was measured after the end of this incubation period. During these 48 h, the catabolism of GHB and the formation of high levels of succinate may have activated the malate-pyruvate shunt impaired by palmitate and generated the antioxidant cofactor NADPH. Alternatively, GHB may have activated the PPS. The high levels of NADH generated by the oxidation of succinic semialdehyde may activate nicotinamide nucleotide transhydrogenase to form additional NADPH [27], however as stated previously, this enzyme is lacking in C57BL/6J mice. Once generated, NADPH may have affected insulin secretion by its antioxidant mechanism as well as by acting on exocytosis and Kv channels, as described earlier.

Although the catabolism of GHB has been shown to generate NADH and succinate in the brain and in peripheral tissues, past studies have demonstrated that GHB only increases oxygen consumption in slices of cerebral grey matter [8,26]. GHB was not able to increase oxygen consumption in cerebral homogenates or in liver slices and, in our study, it was not able to increase oxygen consumption in isolated islets. Oxygen consumption actually tended to be depressed by GHB alone, which could be explained by the diversion of α-ketoglutarate to D-α-hydroxyglutarate in the Krebs cycle. The generation of ROS in response to palmitate may account for the reduction in oxygen consumption via mitochondrial dysfunction, and GHB appeared to prevent this reduction presumably via its antioxidant effect. The effect of GHB to improve mitochondrial function in the presence of palmitate was, however, not significant and was minor compared to its effect on palmitate-induced ROS, which suggests that other mechanisms related to GHB’s antioxidant effect or to the NADPH elevation contributed to the GHB effect on GSIS.

GHB has been shown to have widespread cellular protective effects [3,4]. Protection against ischemic reperfusion injury has been demonstrated in the brain, gut, liver, and vascular endothelium. Thus, in addition to protecting the ß-cells of the pancreas from the toxic effects of high levels of NEFA in diabetes, GHB may also be able to protect the vascular endothelium in diabetes from the high levels of oxidative stress and vascular damage that leads in time to major end organ failure [28,29]. Long-acting formulations of GHB are now being developed for once nightly use (Avadel Pharmaceuticals, Dublin, Ireland; XW Pharma Inc., Redwood City, CA, USA) This may facilitate its potential use as an adjuvant in patients with diabetes and sleep disorders. Stress and poor sleep have long been recognized to interfere with the optimum control of the disease. Insufficient sleep is recognized to alter the regulation of energy intake and circadian timing and to lead to weight gain, insulin resistance, and glucose intolerance [30]. It might also be possible to prevent the central effects of GHB by PEGylating the molecule, which impedes the blood-brain barrier passage, as recently shown with other compounds [31,32], while retaining its favorable peripheral antidiabetic effects.

## 5. Conclusions

GHB is shown to be effective at quenching the generation of ROS in response to the NEFA palmitate, and in preventing the impairment of GSIS normally caused by palmitate. GHB’s protective actions may be related to its capacity to stimulate the formation of the antioxidant cofactor NADPH.

## Figures and Tables

**Figure 1 cells-11-00545-f001:**
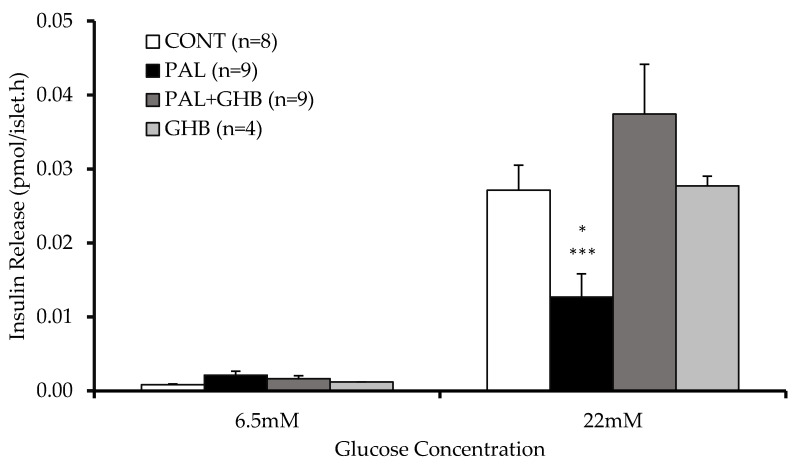
Effects of palmitate and gamma-hydroxybutyric acid (GHB) on glucose-stimulated insulin secretion in mouse islets. Islets were treated for 48 hrs with (1) control media, (2) palmitate (0.4 mM), (3) palmitate (0.4 mM) and GHB (5 mM), and (4) GHB (5 mM) alone. GSIS was performed after 1 h recovery. Palmitate treatment impaired GSIS while GHB restored GSIS in palmitate treatment. GHB by itself did not affect GSIS. Data are means ± SE. One-way nonparametric ANOVA on ranks for repeated measurements followed by Tukey’s test was performed to compare differences between treatments. * *p* < 0.05 vs. all, *** *p* < 0.001 vs. PAL + GHB.

**Figure 2 cells-11-00545-f002:**
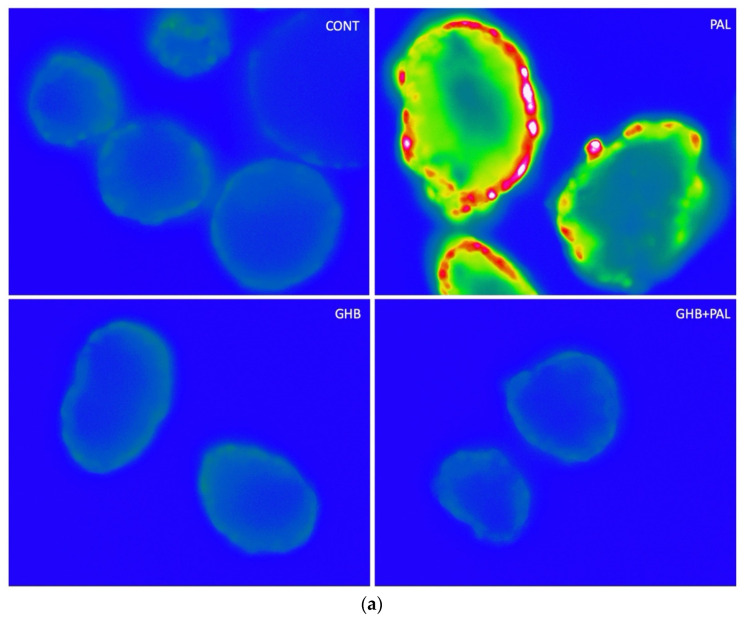
Effects of palmitate and gamma-hydroxybutyric acid (GHB) on ROS levels in mouse islets. ROS levels were determined by H2DCF-DA. Islets were treated for 48 hrs with (1) control media, (2) palmitate (0.4 mM), (3) palmitate (0.4 mM) and GHB (5 mM), and (4) GHB (5 mM) alone. Approximately 10 islets were measured per experiment (*n*). (**a**) Representative images. (**b**) Quantification of results. Palmitate treatment increased ROS while GHB had a protective effect. GHB alone did not affect H2DCF-DA-detected ROS. RFI = relative fluorescence intensity. Data are means ± SE. One-way parametric ANOVA followed by Tukey’s test was performed to compare differences between treatments. * *p* < 0.05 vs. all, ** *p* < 0.01 vs. PAL + GHB.

**Figure 3 cells-11-00545-f003:**
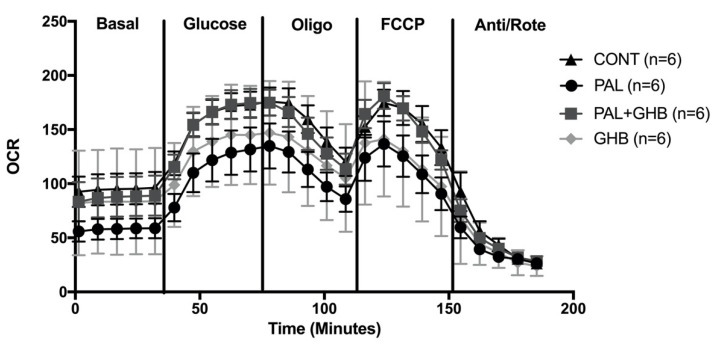
Effects of palmitate and gamma-hydroxybutyric acid (GHB) on oxygen consumption rate in mouse islets. Measurement of oxygen consumption rate (OCR) with Seahorse XF cell Mito Stress Test (Seahorse Bioscience, Billerica, MA, USA). Islets were treated for 48 hrs with (1) control media (black triangle), (2) Palmitate (0.4 mM) (black circle), (3) Palmitate (0.4 mM) and GHB (5 mM) (dark-gray square), and (4) GHB (5 mM) alone (light-gray diamond). Palmitate treatment tended to reduce OCR while GHB seemed to have a protective effect. Data are means ± SE. One-way parametric ANOVA for repeated measures followed by Tukey’s test was performed to compare differences between treatments.

**Figure 4 cells-11-00545-f004:**
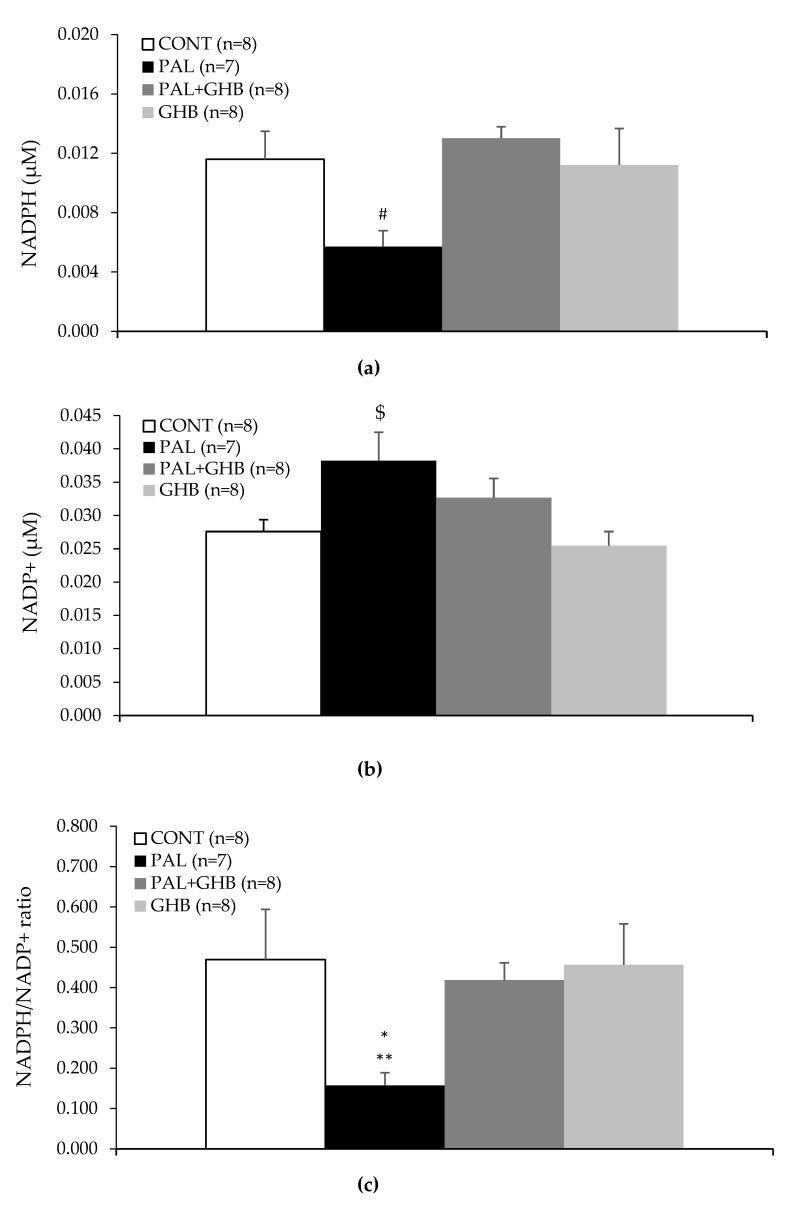
Effects of palmitate and gamma-hydroxybutyric acid (GHB) on NADPH and NADP+ in mouse islets. Islets were treated for 48 hrs with (1) control media, (2) Palmitate (0.4 mM), (3) Palmitate (0.4 mM) and GHB (5 mM), and (4) GHB (5 mM) alone. (**a**) NADPH measurement in islets. (**b**) NADP+ measurement in islets. (**c**) NADPH/NADP+ ratio in islets. Palmitate treatment reduced the NADPH/NADP+ ratio in islets while treatment with GHB reversed this effect. Palmitate tended to reduce NADPH levels and increase NADP+ in islets. Data are means ± SE. One-way parametric ANOVA followed by Tukey’s test was performed to compare differences between treatments for NADP+. One-way nonparametric ANOVA on ranks followed by Tukey’s test was performed to compare differences between treatments for NADPH and NADPH/NADP+ ratio. * *p* < 0.05 vs. ALL; ** *p* < 0.01 vs. PAL + GHB and GHB; # *p* < 0.01 vs. PAL + GHB; $ *p* < 0.05 vs. GHB.

## Data Availability

The data presented in this study are available on request to the corresponding author.

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
