# Peer review of "Prevention of Lipotoxicity in Pancreatic Islets with Gammahydroxybutyrate"

_cells, 2022, doi:10.3390/cells11030545_

Round 1

Reviewer 1 Report

This an interesting experimental research addressing the possible therapeutic solutions for lipotoxicity- induced beta cell dysfunction in type 2 diabetes. The concept of this article is interesting and the experiments were well performed, but there are some minor points for revision.

  1. Introduction: the authors hypothesize that following the obtained results, GHB or related compounds could play a role as adjuvants "in the treatment of type 2 diabetes in patients with sleep disorders." The phrase should be checked and clarified, it probably refers to treating type 2 diabetes in general, the experiment was not designed to test sleep disorders.
  2. DiscussionsThe interpretation that GHB is as effective as the classical anti-oxidants, (N-acetylcysteine, taurine, and tempol) in blocking the formation of ROS doesn't seem well-justified because no direct comparison between GHB and other antioxidants was tested in this experiment.
  3. The Conclusions should be also reviewed for the same reason.

Author Response

Responses to reviewers

We thank the reviewers for their helpful comments (in italics). Our responses are in red.

Reviewer #1

This an interesting experimental research addressing the possible therapeutic solutions for lipotoxicity- induced beta cell dysfunction in type 2 diabetes. The concept of this article is interesting and the experiments were well performed, but there are some minor points for revision.

  1. Introduction: the authors hypothesize that following the obtained results, GHB or related compounds could play a role as adjuvants "in the treatment of type 2 diabetes in patients with sleep disorders." The phrase should be checked and clarified, it probably refers to treating type 2 diabetes in general, the experiment was not designed to test sleep disorders.

We have deleted “in patients with sleep disorders” in the last sentence of the Introduction.

  1. Discussions: The interpretation that GHB is as effective as the classical anti-oxidants, (N-acetylcysteine, taurine, and tempol) in blocking the formation of ROS doesn't seem well-justified because no direct comparison between GHB and other antioxidants was tested in this experiment.

The reviewer is correct and we have modified the sentence to eliminate the comparison with classical antioxidants (see line 277).

  1. The Conclusions should be also reviewed for the same reason.

The conclusion has also been reviewed (see line 352).

Reviewer 2 Report

It is necessary to include more details in the handling of mice. Eg Cage space in each animal, type of previous feeding before slaughter, slaughter technique (anesthesia), collection of samples.

Why did the authors run a nonparametric statistical analysis? . They performed a test of homogeneity of variance. Generally the tukey test is only used in parametric statistics. Authors must perform a detailed statistical analysis.

Author Response

Responses to reviewers

We thank the reviewers for their helpful comments (in italics). Our responses are in red.

Reviewer #2

It is necessary to include more details in the handling of mice. Eg Cage space in each animal, type of previous feeding before slaughter, slaughter technique (anesthesia), collection of samples.

We have added the requested information (see lines 74-75 and 77-82).

Why did the authors run a nonparametric statistical analysis? . They performed a test of homogeneity of variance. Generally the tukey test is only used in parametric statistics. Authors must perform a detailed statistical analysis.

As now described in the paper on lines 128-132, we used a nonparametric analysis because most parameters (actually all except ROS, OCR and NADP+) were not normally distributed according to the Shapiro-Wilk test. We therefore used ANOVA after rank transformation which allowed us to use Tukey’s test. For the parameters that were normally distributed, we conducted a parametric analysis with ordinary ANOVA and Tukey’s test. We have also added the statistical analysis used in each figure legend.

Reviewer 3 Report

The manuscript reads well.  The methods are  clearly described and the results are clearly presented. The conclusions are based on the results.

Author Response

We thank the reviewers for their helpful comments (in italics). Our responses are in red.

Reviewer #3

The manuscript reads well.  The methods are clearly described and the results are clearly presented. The conclusions are based on the results.

We thank the reviewer for their comments.